# First Report of the Yezo Virus Isolates Detection in Russia

**DOI:** 10.3390/v17081125

**Published:** 2025-08-15

**Authors:** Mikhail Kartashov, Kirill Svirin, Alina Zheleznova, Alexey Yanshin, Nikita Radchenko, Valentina Kurushina, Tatyana Tregubchak, Lada Maksimenko, Mariya Sivay, Vladimir Ternovoi, Alexander Agafonov, Anastasia Gladysheva

**Affiliations:** 1State Research Center of Virology and Biotechnology “Vector”, 630559 Kol’tsovo, Russia; kartashov_myu@vector.nsc.ru (M.K.); svirin_ka@vector.nsc.ru (K.S.); zheleznova_as@vector.nsc.ru (A.Z.); yanshin_ao@vector.nsc.ru (A.Y.); radchenko_ns@vector.nsc.ru (N.R.); kurushina_vyu@vector.nsc.ru (V.K.); maksimenko_lv@vector.nsc.ru (L.M.); sivay_mv@vector.nsc.ru (M.S.); tern@vector.nsc.ru (V.T.); agafonov@vector.nsc.ru (A.A.); 2Physics Department, Novosibirsk State University, 630090 Novosibirsk, Russia; 3Natural Sciences Department, Novosibirsk State University, 630090 Novosibirsk, Russia

**Keywords:** ticks, *Ixodes persulcatus*, tick-borne diseases, *Orthonairovirus*, Yezo virus, protein structure, nucleoprotein, AlphaFold

## Abstract

The recent discovery of the Yezo virus (YEZV) in Japan and China has raised particular concern due to its potential to cause human diseases ranging from mild febrile illnesses to severe neurological disorders. We report, for the first time, the detection of five YEZV isolates in *I. persulcatus* ticks from three regions of Russia. The analysis was performed using 5318 ticks of two *Ixodes* genus collected in 2024 from 23 regions of Russia. The minimum infection rate of YEZV in Russia among *I. persulcatus* ticks was 0.12% (95% CI: 0.05–0.28). The westernmost and northernmost YEZV detection points have been recorded. YEZV isolates circulating in Russia are genetically diverse. Protein domains of Russian YEZV isolates’ genomes were characterized using HMMER, AlphaFold 3, and InterProScan. The YEZV nucleoprotein (N) of Russian isolates has a racket-shaped structure with “head” and “stalk” domains similar to those of *Orthonairovirus haemorrhagiae*. The Lys261–Arg261 substitution in the YEZV N Chita 2024-1 isolate occurs in the α11 structure in the region of interaction with viral RNA. Our results show that the distribution area of YEZV is much wider than previously known, provide new data on complete YEZV genomes, extend our structural insight into YEZV N, and suggest a potential target for antiviral drug development to treat YEZV infection.

## 1. Introduction

The unfavorable epidemiological situation with tick-borne infections may be characterized not only by an increase in the incidence of already known diseases but also by the identification of new nosological forms and pathogens, the role of which remains poorly understood. Over the past decades, primarily due to the method of massive parallel sequencing, many novel tick-borne viruses have been described to cause diseases in humans or whose pathogenicity has not yet been studied [1]. An emerging tick-borne viral disease includes the novel tick-borne virus, the *Orthonairovirus yezoense* (Yezo virus, YEZV), belonging to the genus *Orthonairovirus* in the family *Nairoviridae*.

The YEZV viral genome is composed of three single-stranded negative-sense RNA (ssRNA(−)) segments referred to as small (S) (~1.6 kb), medium (M) (~4 kb), and large (L) (~12.1 kb) segments which encode viral nucleoprotein (N), the glycoprotein precursor (GPC), and L-protein, respectively, all of which have complementary 3′ and 5′ termini [2].

YEZV was first detected in two tick-bite patients with acute febrile illness, including leukopenia and thrombocytopenia in Hokkaido, Japan. Retrospective studies of serum samples from 248 patients with suspected tick-borne disease in Hokkaido since 2014 have detected YEZV RNA in five additional individuals [3]. YEZV infection has also been reported in China. The first case was reported in a study of 402 tick-bite patients from northeast China, which showed the presence of YEZV RNA in one patient [4]. Later, 18 patients with YEZV infection were identified among 988 participants from the Mudanjiang Central Forest Hospital, Heilongjiang Province, in northeast China [5]. YEZV RNA was detected in a large study of ticks: *Ixodes persulcatus* (*I. persulcatus*), *Ixodes ovatus*, *Haemaphysalis megaspinosa*, *Dermacentor nuttalli*, *Rhipicephalus microplus*, including *I. persulcatus* taken from migratory birds (black-faced buntings, *Emberiza spodocephala*) [4,5,6,7]. Serological tests have demonstrated that YEZV infection is common among wild animals. Antibodies to YEZV have been detected in serum samples from raccoons (*Procyon lotor*) and raccoon dogs (*Nyctereutes procyonoides albus*) in Hokkaido [3].

It should be noted that infections caused by tick-borne orthonairoviruses can cause serious diseases in humans and are becoming a global public health problem [8]. Thus, the aim of this research was to search for YEZV RNA in ticks from various regions of Russia, followed by the identification and analysis of full-length sequences of the identified YEZV isolates.

## 2. Materials and Methods

### 2.1. Collection and Processing of Ticks

Ticks were collected by flagging from vegetation in the 23 regions of the Russian Federation during the summer of 2024 (Figure 1 and Appendix A). Ticks were identified using taxonomic keys. The study analyzed 4122 individual *I. persulcatus* adult ticks and 1196 individual *I. ricinus* adult ticks.

Ticks were washed twice with 70% ethanol to remove external contaminants and external microflora and then stored at −20 °C until further studies. Homogenization of the ticks was carried out using the laboratory homogeniser TissueLyser II (QIAGEN, Hilden, Germany) in 300 μL 0.9% saline solution. Homogenates were centrifuged, supernatants were collected and stored at −80 °C. Viral RNA from 150 μL tick supernatants was isolated with ExtractRNA (Evrogen, Moscow Russia), according to the manufacturer’s protocols. Reverse transcription was performed with random hexamer primer (R6) and MMLV reverse transcriptase (Evrogen, Russia), according to the manufacturer’s protocols.

### 2.2. RT-PCR Screening

Screening of the tested samples for YEZV RNA detection was performed by real-time polymerase chain reaction (qPCR) using screening primers complementary to the fragment of the M segment: YEZV_F CACCAGGCATTTACCTCTACTT, YEZV_R TGGAGTCAAGGGCTGTTATG, and YEZV_Z CY5-TGCCAGGGCTACTGTGATGCATAA-BQ2 [9]. qPCR was performed on a CFX96 Touch thermocycler (Bio-Rad, Hercules, CA, USA) in 25 μL of BioMaster HS-Taq PCR reaction mix (Biolabmix, Novosibirsk, Russia) containing 0.4 pM of each primer and 0.2 pM of probe under the following conditions: polymerase activation at 95 °C for 5 min and then 45 cycles: 95 °C for 10 s, 55 °C for 20 s, 68 °C for 20 s.

### 2.3. Next Generation Sequencing (NGS)

To increase the amount of target YEZV RNA with positive samples, targeting PCR was performed using a panel of primers we developed for this study (Appendix A). Phusion Flash High-Fidelity PCR Master Mix (Thermo Fisher Scientific, Waltham, MA, USA) containing polymerase with high fidelity and processivity was used for targeting PCR.

The resulting dsDNA fragments were purified from unspent components and reaction products with AMPure beads (Beckman Coulter, Brea, CA, USA) and then used to prepare NGS libraries. Nucleic acid concentrations were measured with Qubit 2.0 using the Qubit dsDNA HS Assay Kit (Thermo Fisher Scientific, MA, USA). The NGS library preparation was carried out using the NEBNext Ultra II FS DNA Library Prep Kit for Illumina (NEB, Hitchin, UK), which performs fragmentation, end repair and dA-tailing, and adapter ligation with a single enzyme mix. The sequencing was performed on Illumina MiSeq platform (Illumina, San Diego, CA, USA).

### 2.4. NGS Data Analysis

To remove adapters, short sequences, and low-quality sequences, FASTQ files were processed using fastp v0.20.1 with a quality score less than 20 and a length less than 30 nucleotides, https://github.com/OpenGene/fastp/ (accessed on 10 February 2025) [10]. The preprocessed reads were aligned to the YEZV reference genome (GenBank assembly numbers: LC790676, LC790675, LC790674), obtained from the NCBI GenBank database, using BWA MEM v0.7.18, https://github.com/lh3/bwa (accessed on 10 February 2025) [11]. SAM/BAM file processing and analysis was performed with Samtools v1.11 [12]. The iVar v1.2.2 program was used to extract the consensus sequence from the BAM files [13].

### 2.5. Sequence Alignments and Phylogenetic Reconstruction

All available complete genome sequences were downloaded for YEZV (*n* = 33) from NCBI (https://www.ncbi.nlm.nih.gov/labs/virus), accessed on 10 February 2025 and their sequence information was recorded. Multiple sequence alignment was performed using MEGA X (PSU, Philadelphia, PA, USA). These programs were also used to calculate sequence identities. A custom Python script was developed using the Biopython library to compute pairwise sequence identity matrices (https://github.com/AlexeiYanshin/heatmap/blob/main/heatmap_pairwise_alignment.ipynb (accessed on 24 July 2025)). The analysis was performed using Python v3.12.7 with the following libraries: Biopython v1.85 for sequence parsing and manipulation, Pandas v2.2.3 for data handling, Seaborn v0.12.2, and Matplotlib v3.7.2 for visualization.

Maximum likelihood trees were constructed using IQ-TREE software v2.4.0 (Australian National University, Canberra, Australia). The resulting phylogenetic tree was visualized using iTOL v7.1, https://itol.embl.de/, accessed on 27 March 2025 [14].

### 2.6. Nucleotide Sequence Accession Numbers

Nucleotide sequences determined in the study (*n* = 5) are available in the GenBank database under accession numbers: PV061568-PV061572 for YEZV L segment, PV061573-PV061577 for YEZV M segment, and PV061578-PV061582 for YEZV S segment.

### 2.7. Statistical Analysis

The minimum YEZV infection rate was estimated using 95% confidence intervals (95% CI). The 95% confidence interval was calculated using Wilson’s estimator without correction for continuity (https://pedro.org.au/wp-content/uploads/CIcalculator.xls (accessed on 10 February 2025)).

### 2.8. Functional Annotation of Viral Proteins

Functional annotation of viral proteins encoded by the YEZV M and YEZV L segments was performed using HMMER v3.1b2 (University of California, Santa Cruz, CA, USA), https://www.ebi.ac.uk/Tools/hmmer/ (accessed on 1 April 2025) [15] and InterProScan v104.0 (EMBL-EBI, Cambridgeshire, UK), https://www.ebi.ac.uk/interpro/ (accessed on 1 April 2025) [16]. Analysis of the full-length YEZV L-protein using InterProScan and HMMER v3.1b2 failed to directly localize the endonuclease domain. We approximated the position of the endonuclease domain in the Russian YEZV isolates genome based on recently published data [17].

Spatial structure models of the YEZV endonuclease domain, YEZV OTU-like domain and YEZV N were performed using the AlphaFold 3 server (Google DeepMind, London, UK), https://alphafoldserver.com/welcome (accessed on 1 April 2025) [18] and validated against known orthonairoviruses domains using the FoldSeek server (Seoul National University, Seoul, South Korea), https://search.foldseek.com/ (accessed on 1 April 2025) [19]. Structural models were selected based on the confidence coefficient for each amino acid with allowance for an AlphaFold 3 predicted local distance difference (pLDDT) scaled from 0 to 100, which estimates the difference in Cα interatomic distances between the reference and the predicted structures. Pairwise alignment of the spatial structures of viral proteins and generated spatial structure models of YEZV proteins was performed using a Pairwise Structure Alignment tool (Research Collaboratory for Structural Bioinformatics Protein Data Bank, RCSB PDB, Piscataway, NJ, USA), https://www.rcsb.org/alignment/ (accessed on 1 March 2025) with TM-align for pairwise structural alignment [20]. The level of topological similarity was evaluated based on the root mean square deviation (RMSD) coefficient and assessment of the number of superimposed atoms in structures (TM-score) on a scale from 0 to 1, where 1 indicates a perfect match between the predicted model and the reference structure. Tertiary structure models of viral proteins were visualized using UCSF ChimeraX v1.15rc (University of California, San Francisco, CA, USA) [21].

Pairwise identity alignment of secondary structures of YEZV viral proteins were generated with an ESpript v3.0. 0 (Institute for the Biology and Chemistry of Proteins, Lyon, France), https://espript.ibcp.fr/ (accessed on 1 April 2025) [22].

The signal peptide, transmembrane, intramembrane, and extramembrane domains were localized with InterProScan and SignalP v5.0 (Department of Health Technology, Lyngby, Denmark), https://services.healthtech.dtu.dk/services/SignalP-6.0/ (accessed on 1 April 2025) [23].

## 3. Results

### 3.1. Tick Identification and Yezo Virus Molecular Screening

We analyzed 4122 *I. persulcatus* ticks, collected in 20 regions of Russia and 1196 *I. ricinus* ticks, collected in 3 regions of Russia (Table 1 and Appendix A). Screening for YEZV RNA revealed 5 positive *I. persulcatus* ticks, while YEZV RNA was not detected in *I. ricinus* ticks (Table 1 and Appendix A). As expected, the largest number of YEZV-positive samples was found in the Primorsky territory: *Orthonairovirus yezoense* isolate Primorye 2024-1, *Orthonairovirus yezoense* isolate Primorye 2024-2, *Orthonairovirus yezoense* isolate Primorye 2024-3. Two YEZV-positive ticks were caught on a flag in the Dukhovskaya River valley of the Primorsky territory (geographic coordinates 44.1704174; 131.6892353 and 44.5058746; 131.5328941) and one YEZV-positive tick in the Talovka River valley of the Primorsky territory (geographic coordinates 45.4984781; 134.2434771). The YEZV tick minimum infection rate of ticks in the Primorsky territory was 0.8% (95% CI: 0.3–2.4). One YEZV-positive tick (*Orthonairovirus yezoense* isolate Khabarovsk 2024-1) was detected in the Khabarovsk territory (geographic coordinates 47.24863; 134.39577), the YEZV minimum infection rate of ticks in this region was 1.3% (95% CI: 0.2–6.8). The discovery of the YEZV-positive tick (*Orthonairovirus yezoense* isolate Chita 2024-1) in the Alkhanai National Park, Duldurginsky district of the Zabaikalsky territory (geographic coordinates 50.9242852; 113.2820464) was unexpected. The YEZV minimum infection rate of ticks in this region was 0.7% (95% CI: 0.1–3.6). To date, this is the westernmost and northernmost point of YEZV detection.

### 3.2. Yezo Virus Genome

The complete coding sequences of five Russian YEZV isolates with partially sequenced 3′-5′ UTR RNA regions obtained in this study were placed in GenBank under numbers PV061568–PV061582. The S YEZV segment has a length of 1606 nucleotides (Open reading frame (ORF)—1509 nucleotides), the M segment—4227 nucleotides (ORF—4071 nucleotides), and the L segment—12,089 nucleotides (ORF—11,817 nucleotides). As in previously discovered YEZV, the S segment of Russian YEZV isolates encodes a nucleoprotein (N; 502 aa), the M segment encodes a glycoprotein precursor complex (GPC-protein; 1356 aa), and the L segment encodes a multi-functional protein complex (L-protein; 3938 aa), including an RNA-dependent RNA polymerase (RdRp) (Figure 2).

The S segment of the Russian YEZV isolates encodes the full-length N, and according to previous studies on the function of nucleoproteins from other families, some of these N may be able to recognize specific viral RNA sequences but mostly bind to viral ssRNA(−) in a nonspecific way (Appendix A) [24]. The Russian YEZV isolates GPC-protein contains an N-terminal signal peptide (from 1 aa to 21 aa) and post-translationally cleavable mature glycoproteins: Gn (from 369 aa to 687 aa) and Gc (from 777 aa to 1319 aa) necessary for orthonairoviruses to recognize host receptors. Three transmembrane helices were detected in the Russian YEZV isolates GPC-protein (Figure 2). Putative YEZV Gn has two non-cytoplasmic regions (369–537 aa and 686–687 aa), two transmembrane helices (538–567 aa and 668–665 aa), and one cytoplasmic region (568–667 aa). Putative YEZV Gc has one non-cytoplasmic regions (777–1286 aa), one transmembrane helix (1287–1315 aa), and one cytoplasmic region (1316–1319 aa). The Russian YEZV isolates L-protein contains the viral homolog of the ovarian tumor protease domain (OTU-like domain) at the N-terminus from 8 aa to 160 aa, putative endonuclease domain from 673 aa to 883 aa, and the functional core of the RdRp from 2260 aa to 2570 aa with high conservative RdRp motifs: motif A “DNTKWG” from 2348 to 2353 aa, motif B “QGIHHATSS” from 2463 to 2471 aa, motif C “GSSDDY” from 2504 to 2509 aa, motif D “CQMKDSAKTL” from 2549 to 2558 aa, motif E “EFYSEFM from 2565 to 2571 aa, and motif F “KAQLGG” from 2261 to 2266 aa (Figure 2 and Appendix A). We identified YEZV OTU-like domain and YEZV putative endonuclease domain in all Russian YEZV isolates in identical locations in each species (Appendix A, Appendix A). The OTU-like domain has the potential to modify antiviral type I interferon responses and influence the severity of infectious disease caused by orthonairoviruses [6]. Putative endonuclease domain is responsible for cap-snatching endonuclease activity [25].

### 3.3. Homology Analysis of YEZV Nucleotide and Amino Acid Sequences

Nucleotide sequence alignment of YEZV segments revealed high and nearly identical levels of similarity between related genes of the five Russian YEZV isolates analyzed (L = 97.4–99.9%, M = 97.7–100%, S = 97.2–100%) (Figure 3 and Appendix A). The level of amino acid sequence differences for all three Russian YEZV isolates segments is approximately 0.5%. Compared with other YEZV isolates previously detected in Japan and China, Russian YEZV isolates have a nucleotide sequence identity level of 95.4–99.8% for the L segment, 95.3–99.7% for the M segment, and 93.4–100% for the S segment (Figure 3 and Appendix A). Meanwhile the level of amino acid sequence identity is 99.3–99.7% for the L segment, 97.4–99.7% for the M segment, and 99.4–100% for the S segment. BLASTx search of the Russian YEZV isolates S segment revealed the highest similarity to *Orthonairovirus sulinaense* (Sulina virus) isolate b30 (GenBank ID: PP260004), isolated from the *I. ricinus* tick in May 2022, Latvia [26]. The identity level was 66.64% with 79% coverage.

### 3.4. Phylogenetic Analysis of YEZV Genomic Segments

Phylogenetic analysis was performed using YEZV ORF nucleotide sequences from the S, M, and L segments to investigate the evolutionary relationships and genetic diversity of the five Russian YEZV isolates relative to previously known YEZV isolates with complete cds available at GenBank (*n* = 33). There is no clear clustering of known YEZV isolates by geographic location or host. According to the phylogenetic tree for YEZV S segment, Russian YEZV isolates are divided into two main YEZV clades. YEZV Khabarovsk 2024-1 isolate is included in one of them (blue clade), and YEZV Primorye 2024-1, YEZV Primorye 2024-2, YEZV Primorye 2024-3, and YEZV Chita 2024-1 isolates are included in the other (yellow clade) (Figure 4a). While according to the phylogenetic trees for YEZV M segment and YEZV L segment, a division of Russian YEZV isolates into four clades is observed. YEZV Primorye 2024-1 and YEZV Primorye 2024-2 isolates together form one clade (red clade). YEZV Khabarovsk 2024-1 (blue clade), YEZV Primorye 2024-3 (purple clade), and YEZV Chita 2024-1 (yellow clade) isolates form their own separate clades (Figure 4b,c). YEZV Chita 2024-1 isolate clusters with Japanese YEZV isolates associated with acute febrile illness, recovered from tick-bite patients.

### 3.5. Nucleoprotein Structure Comparison of the Russian Yezo Virus Isolates

We modeled the tertiary structures of YEZV N to determine the functional significance of the detected amino acid substitutions between Russian YEZV isolates (Figure 5a). The models of YEZV N tertiary structures with the highest pLDDT value, ranging from 86.7 to 87.4, were selected. Pairwise alignment of these YEZV N tertiary structures showed a high level of structural similarity, with TM-score coefficients varying from 0.93 to 0.99, and the RMSD coefficient varied from 0.85 Å to 2.47 Å (Appendix A). However, when comparing the tertiary structures of YEZV N and the crystal structures of N closely related viruses, significantly lower similarity coefficients were observed (0.71 < TM-score < 0.84, 2.62 Å < RMSD < 3.29 Å). The highest TM-score = 0.84 was found for *Orthonairovirus haemorrhagiae* N (Crimean–Congo hemorrhagic virus, CCHFV, PDB ID: 4AQF) and for *Orthonairovirus hazaraense* N (Hazara virus, HAZV, PDB ID: 5A97).

N YEZV of the Russian YEZV isolates has a two-domain racket-shaped structure typical for nucleoproteins of orthonairoviruses, consisting of a “head” domain and a “stalk” domain [27]. Both the “head” and “stalk” domains are mainly consisting of α-helices (Figure 5a). The N YEZV structures show high flexibility in the “stalk” domain. Significant conformational differences are exhibited in the orientation between the “head” and “stalk” domains (Appendix A). These observations suggest structural flexibility of N YEZV that may provide a basis for switching between alternative conformations of N YEZV during important functions, such as viral RNA binding and oligomerization, to form ribonucleoprotein complexes.

It was found that the Cys to Ser substitutions at position 40 aa (Cys40–Ser40) in YEZV N Chita 2024-1 isolate, Ser145–Tyr145 in the YEZV N Khabarovsk 2024-1 and Primorye 2024-1 isolates, Pro502–His502 in the YEZV N Khabarovsk 2024-1 and Primorye 2024-1 isolates are located in the “head” domain of N YEZV and do not affect regions important for N functioning. However, the Lys261–Arg261 substitution occurs in the α11 structure of YEZV N Chita 2024-1 isolate, presumably in the pocket of interaction with viral RNA [28]. Investigation of the electrostatic potential on the surface of the N YEZV complex structure and the YEZV S segment RNA fragment of 62 nucleotides in size (3′-GAUUGUGGAUCCACUUAUCCAGCUCUUCUCUAGUCUUUGCU_41_UCAAUCAGACGUCCAUCUCCG-5′) showed that YEZV RNA interacts with N YEZV in the positively charged region and confirms the interaction of YEZV RNA Uracil at position 41 with YEZV N at position 261 aa (Figure 5). There is a large positively charged cavity (crevice and platform) located at the center of the N YEZV “head” domain, and a positively charged region in the N YEZV “stalk” domain near to the “head” domain (Figure 5b). The RNA binding region in the YEZV N is formed by a cluster of amino acids: Ser40-Ser145-Arg261-Pro502 (Appendix A). The N binds each RNA segment and forms ribonucleoprotein complexes, the functional template of the L protein RdRp for transcription and replication [17].

## 4. Discussions

In recent years, NGS has been used to identify numerous tick-borne viruses that are pathogenic to humans. As global climate conditions change and humans encroach on previously untouched natural areas, the likelihood that tick-borne diseases will become more prevalent is expected to increase [29]. Rising global temperatures allow ticks to colonize previously inhospitable areas, increasing their distribution and accelerating their life cycle. This results in longer periods of tick activity, increasing the likelihood of tick-borne viral infections being transmitted to humans as people and animals are exposed to potential tick bites for longer periods of time [30]. Thus, YEZV has emerged as a novel tick-borne threat [8].

We report for the first time the detection of five YEZV isolates in *I. persulcatus* ticks from three regions of Russia. The minimum infection rate of YEZV in Russia among *I. persulcatus* ticks was 0.12% (95% CI: 0.05–0.28). The main carriers of the most common transmissible human diseases of both viral and bacterial origin in the Asian part of Russia are *I. persulcatus* ticks. The *I. persulcatus* tick (“Taiga tick”) has high ecological plasticity, which is manifested in the ability to exist in various types of forests of the temperate zone of Eurasia. The most severe epidemiological situation with tick-borne infections is traditionally characteristic of Siberia and the Far East region of Russia, including the territories bordering China [31,32]. Our data confirm that YEZV is currently predominantly transmitted via *I. persulcatus* ticks and is widely distributed in the Far East region of Russia. Russian YEZV isolates are grouped together with Chinese and Japanese YEZV isolates. On the one hand, we do not observe clustering of Russian YEZV isolates and previously known YEZV isolates into phylogroups based on geographic location or host. But on the other hand, YEZV Chita 2024-1 isolate predominantly clusters with Japanese YEZV isolates associated with acute febrile illness. This suggests potential shared ancestry or recent cross-border transmission events, possibly facilitated by migratory birds or overlapping tick vector populations. The division of Russian YEZV isolates into two clades based on the S segment (Figure 4a) contrasts with their separation into four distinct clades in the M and L segment trees (Figure 4b,c). This incongruence may indicate differential evolutionary pressures acting on each genomic segment, with the S segment being more conserved, while the M and L segments exhibit higher variability or host/vector adaptation, where certain genomic regions evolve under different selective constraints depending on transmission cycles. The clustering of Russian tick-derived isolates with Japanese human-pathogenic strains raises important epidemiological questions. Clinical manifestations of the YEZV-infected patient in China were milder (mild lymphocytopenia and mildly increased levels of liver enzymes) than those reported for patients in Japan, where leukopenia, lymphocytopenia, thrombocytopenia, coagulation disorder, and increased levels of liver and heart enzymes have been observed. Therefore, one should probably expect different clinical pictures of YEZV infection in Russia, both milder and more severe. YEZV RNA was also detected in *I. persulcatus* ticks collected from migratory birds [7]. These birds fly to Hokkaido, Japan, from Sakhalin and the Kuril Islands, which also makes it necessary to search for genetic markers of the YEZV in these areas of Russia. The transfer of infected ticks with migratory birds can significantly contribute to the spread of the YEZV range over a greater distance. Finally, due to the extremely limited number of YEZV sequences, our phylodynamic analysis results are inevitably biased and lack evidence for some key nodes. Therefore, more YEZV genomes need to be discovered to better understand the spread and evolution of YEZV.

Protein domains of Russian YEZV isolates were characterized based on the genomic organizations of closely related orthonairoviruses. The L-protein of the Russian YEZV isolates contains at least three functional domains. We found putative YEZV OTU-like domain regulates innate immunity to CCHFV; thus, it may be considered as a potential YEZV virulence factor and should be further explored for therapeutic purposes [6,33]. The YEZV M segment is expected to encode GPC, and the YEZV S segment is expected to encode N. Our results demonstrate high structural similarity of the Russian YEZV N tertiary structure with those of the CCHFV despite the extremely low similarity level of the primary structure. CCHFV is a tick-borne *Orthonairovirus*, which causes severe hemorrhagic fever with a mortality of 30% in more than thirty countries worldwide and is listed as a high-priority pathogen by the World Health Organization [34,35]. N proteins of CCHFV and other ssRNA(−) viruses are key structural and scaffolding components of the viral ribonucleoprotein complex and critical to viral infection [36]. At the same time, the YEZV Chita 2024-1 isolate has an amino acid substitution in the RNA binding region, but this substitution does not affect the binding of the YEZV RNA by the YEZV N. It is likely that we do not observe the effect of amino acid substitution on the YEZV N–RNA structure due to the fact that we consider the YEZV N only in the single-stranded RNA binding mode and do not consider it in the RNA panhandle binding mode formed by the base pairing of complementary nucleotides at the 5′ and 3′ termini of viral genome. The binding of both CCHFV RNA panhandle and single-strand CCHFV RNA induce a conformational change in the CCHFV N, but CCHFV N does not discriminate between viral and non-viral RNA molecules in the single-stranded RNA binding mode [37]. The obtained and analyzed spatial structures of the Russian YEZV N facilitate future discovery of novel therapeutic targets against YEZV infection, as well as strengthen our understanding of YEZV evolution. Recently, immunoinformatic approaches have been employed to design a vaccine targeting YEZV GPC, YEZV RdRp, and YEZV N with enhanced antigenicity and verified non-allergenic and non-toxic properties [38]. However, given the growing global concern about the spread of YEZV and its high structural similarity to CCHFV, there is a need to expand the YEZV monitoring area and our knowledge of the structure and properties of YEZV in order to develop POC-tests, vaccines, and antiviral drugs.

## 5. Conclusions

YEZV poses a significant public health challenge due to increasing susceptibility, underdiagnosis, and lack of awareness. In the current study, we report for the first time the detection of YEZV isolates in Russia. Our results show that the distribution area of YEZV is much wider than previously known, provide new data on the complete YEZV genomes, extend our structural insight into YEZV N, and provide a potential target for antiviral drug development to treat YEZV infection.

## Figures and Tables

**Figure 1 viruses-17-01125-f001:**
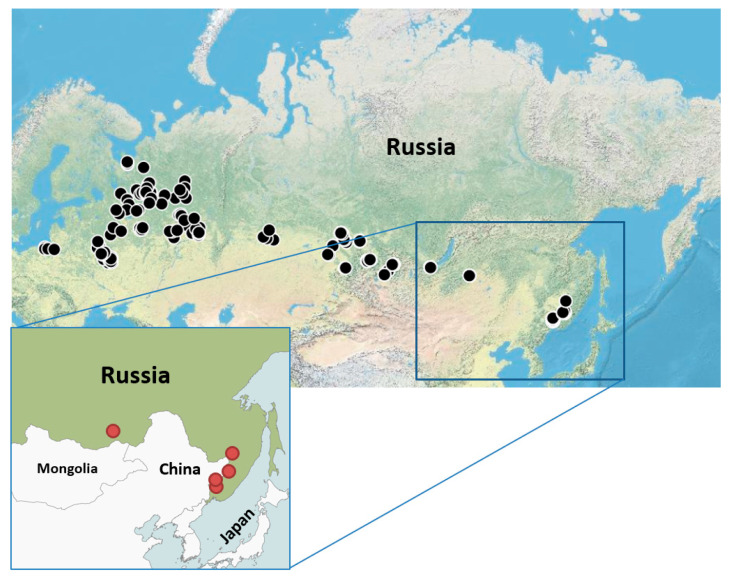
Geographic distribution of tick collections in Russia. Tick collection locations are marked with black circles. YEZV positive tick collection locations are marked with red circles.

**Figure 2 viruses-17-01125-f002:**
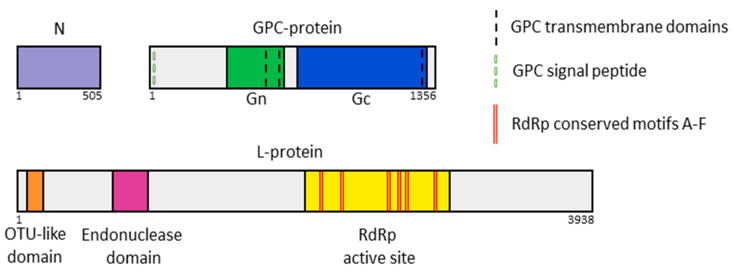
Schematic representation of the coding region of the Russian YEZV isolates genome organization. Putative proteins and domains encoded by the Russian YEZV isolates genome: N (purple), Gn (green), Gc (blue), OTU-like domain (orange), endonuclease domain (purple), and RdRp active site (yellow). In addition, the following are presented: transmembrane domains of GPC-protein (black dotted line), signal peptide of GPC-protein (green dotted line), and conserved motifs of *Nairoviridae* RdRp A-F (red double lines).

**Figure 3 viruses-17-01125-f003:**
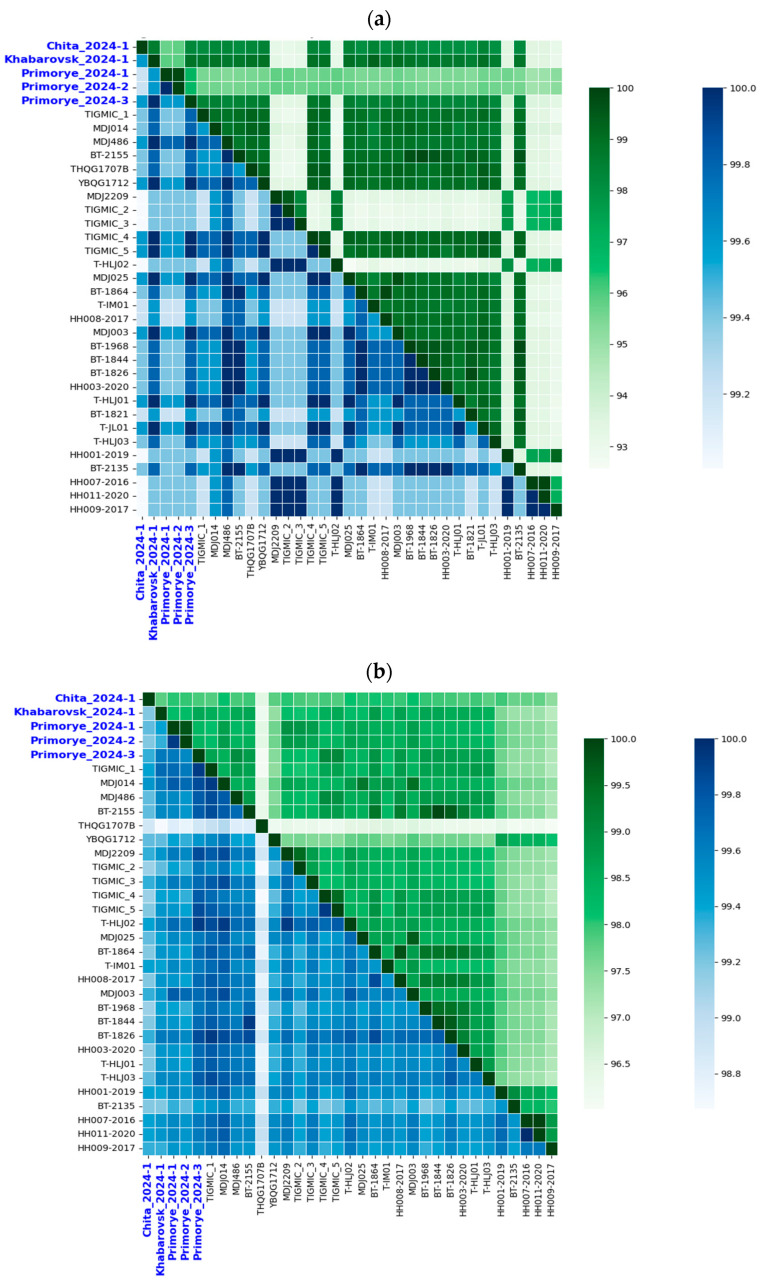
Pairwise sequence identity matrix of YEZV genome segments. (**a**) Identity matrix of YEZV S segment; (**b**) identity matrix of the YEZV M segments; (**c**) identity matrix of YEZV L segments. YEZV sequences are indicated by name of isolate. Russian YEZV isolates are colored blue. Pairwise nucleotide identity matrix is highlighted in green. Pairwise amino acid identity matrix is highlighted in blue.

**Figure 4 viruses-17-01125-f004:**
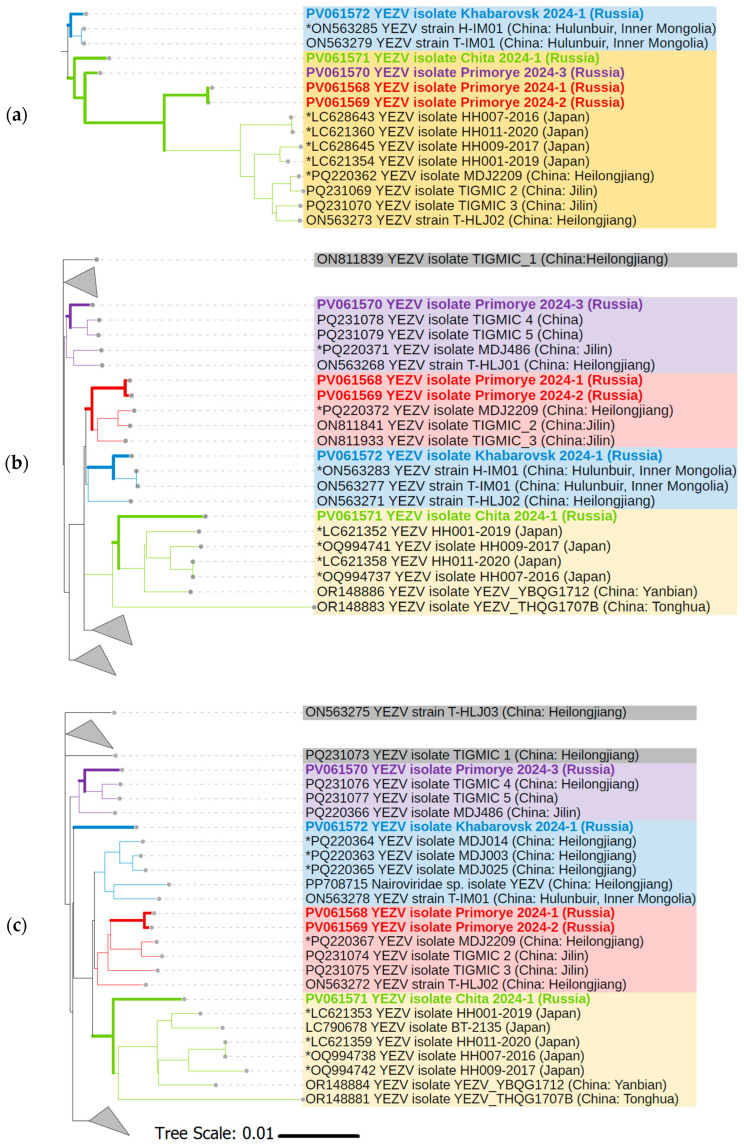
The maximum likelihood consensus tree of the YEZV ORFs: (**a**) S segment; (**b**) M segment; (**c**) L segment. GenBank accession numbers are listed for each strain. Viruses are indicated by GenBank accession number, name and isolate identifier, and location of isolation where available. YEZV sequences obtained in this research are highlighted in bold and different colors. *Homo sapiens* YEZV isolates are marked with an asterisk (*).

**Figure 5 viruses-17-01125-f005:**
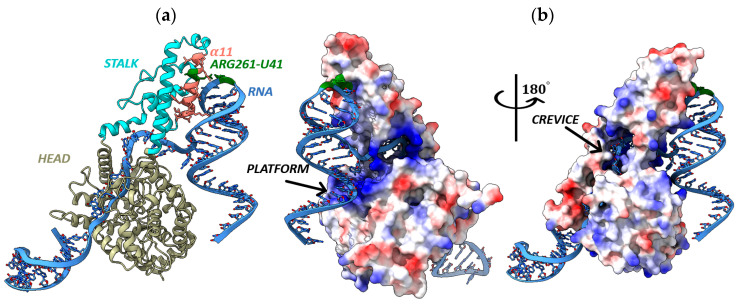
Spatial structure of YEZV N Chita 2024-1 isolate with RNA fragment of YEZV S segment Chita 2024-1 isolate. (**a**) Spatial structure of the complex in ribbon presentation; (**b**) electrostatic surface potential of YEZV N Chita 2024-1 isolate. The positive surface potential is colored blue, and the negative surface potential is colored red. The YEZV N head domain is colored ivory. The YEZV N stalk domain is colored blue. The α11 structure of YEZV N is colored pink, and the region of YEZV N interaction with YEZV RNA is colored green.

**Table 1 viruses-17-01125-t001:** YEZV prevalence in tick species collected in the different regions of Russia.

No. Region	Russia Region	Tick Species	No. Ticks Examined	No. Positive Ticks	Minimum YEZV Infection Rate
1	Primorsky Territory	*I. persulcatus*	368	3	0.8% (95% CI: 0.3–2.4)
2	Khabarovsk Territory	*I. persulcatus*	79	1	1.3% (95% CI: 0.2–6.8)
3	Transbaikal Territory	*I. persulcatus*	150	1	0.7% (95% CI: 0.1–3.6)
4	Other regions	*I. persulcatus*	3525	0	0
		*I. ricinus*	1196	0	0
Total:		*I. persulcatus*	4122	5	0.12% (95% CI: 0.05–0.28)
			5318	5	0.1% (95% CI: 0.04–0.22)

## Data Availability

The original contributions presented in the study have been deposited in the GenBank, accession numbers PV061568 to PV061582, and are included in the article/Appendix A. Further inquiries can be directed to the corresponding author.

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
