# Peer review of "First Report of the Yezo Virus Isolates Detection in Russia"

_viruses, 2025, doi:10.3390/v17081125_

Round 1

Reviewer 1 Report

Comments and Suggestions for Authors

The authors have submitted a report describing five new isolates of Yezo virus from areas of Russia. The complete sequence was determined for phylogenetic analysis and used to investigate protein domains. Overall the manuscript was well presented and I only have minor comments. Please can you add further discussion on the phylogenetic analysis, is their evidence of reassortment for any of these isolates or other Yezo virus isolates. I would suggest amending the title. Referring to case suggests human cases, perhaps replace case with report. Are there any vector competency studies to support proposed vectors?

Comments on the Quality of English Language

The english was suitable standard

Author Response

Thank you for reviewing our manuscript (viruses-3745335) entitled “First case of the Yezo virus isolates detection in Russia” submitted for publication in Viruses.

We thank you for valuable suggestions that allowed us to make the manuscript more convincing and understandable. We accepted your suggestion and made corresponding change in the manuscript. We sincerely appreciate the reviewer’s valuable suggestion regarding the title of our manuscript. Below please find our detailed responses to your questions and comments. All modifications in the manuscript have been highlighted in red.

Point-by-point response to Comments and Suggestions for Authors

Reviewer 1: The authors have submitted a report describing five new isolates of Yezo virus from areas of Russia. The complete sequence was determined for phylogenetic analysis and used to investigate protein domains. Overall the manuscript was well presented and I only have minor comments.

Comments 1: Please can you add further discussion on the phylogenetic analysis, is their evidence of reassortment for any of these isolates or other Yezo virus isolates. 

Response 1: We sincerely appreciate the reviewer's valuable suggestion regarding reassortment analysis in YEZV isolates. In response to your comment, we have now expanded the Discussion section with a dedicated analysis of potential reassortment events (lines 334-347 in the revised manuscript).

Comments 2: I would suggest amending the title. Referring to case suggests human cases, perhaps replace case with report. 

Response 2: As recommended, we have amended the title to replace "case" with "report" to avoid potential ambiguity regarding human cases. The revised title now reads: "First report of the Yezo virus isolates detection in Russia". We believe this adjustment improves the clarity and accuracy of the title.

Comments 3: Are there any vector competency studies to support proposed vectors?

Response 3: In the present study, vector competence experiments were not conducted. However, investigating the potential vectors of Yezo virus and their transmission efficiency is an important direction for our future research. We acknowledge this limitation and plan to address it in subsequent studies.

Reviewer 2 Report

Comments and Suggestions for Authors

In the research article, the authors provide well-demonstrated evidence of the first occurrence of Yezo virus in Russia. The tick-borne disease is becoming increasingly concerning due to climate change, mild winters, and this work presents significant evidence of the expansion of this new tick-borne virus.

Regarding the formatting, structure, and style of the writing, the paper is of very good quality. I noticed only a few typos (such as spaces), but nothing that would decrease the overall quality of the work. What interests me very much is whether the authors collected ticks only from vegetation or also from, for example, migratory birds or rodents. These are usually very important for monitoring the prevalence of pathogens as well; however, this is just a personal interest. Also, did you analyse only the Yezo virus, or were other pathogens included? It could be very interesting to determine if co-infections occur within the ticks (again, just a personal interest). However, there are some comments which could help improve the work.

MAJOR comment:

-Figure 2 can be improved – does the representation of genome organisation only cover the coding regions? Because the figure shows three parts, 505, 1356, and 3938 bp, but according to the data in paragraph lines 187-195, the sizes do not match. Please provide a hint to the reader on how to interpret this figure.

Minor comments:

- Figure S6 has really bad quality, please upload it in a better resolution

- When it is possible, please add the Python script in the GIT hub (line 114), as it is also part of the method.

Author Response

Thank you for reviewing our manuscript (viruses-3745335) entitled “First case of the Yezo virus isolates detection in Russia” submitted for publication in Viruses.

We thank you for valuable suggestions that allowed us to make the manuscript more convincing and understandable. We accepted your suggestion and made corresponding change in the manuscript. We minor revised our article. Below please find our detailed responses to your questions and comments. All modifications in the manuscript have been highlighted in red.

Point-by-point response to Comments and Suggestions for Authors

Reviewer 2: In the research article, the authors provide well-demonstrated evidence of the first occurrence of Yezo virus in Russia. The tick-borne disease is becoming increasingly concerning due to climate change, mild winters, and this work presents significant evidence of the expansion of this new tick-borne virus.

Regarding the formatting, structure, and style of the writing, the paper is of very good quality. I noticed only a few typos (such as spaces), but nothing that would decrease the overall quality of the work.

Comments 1: What interests me very much is whether the authors collected ticks only from vegetation or also from, for example, migratory birds or rodents. These are usually very important for monitoring the prevalence of pathogens as well; however, this is just a personal interest.

Response 1: We appreciate the reviewer’s interest in the sampling methodology. In this study, ticks were collected exclusively from vegetation using a standard flagging technique. Specifically, a white waffle cloth flag (60×100 cm) was used for sampling across various habitats, including open areas (low-grass meadows, steppe biotopes) as well as meadow and forested areas with tall grass and shrubs. The collected ticks were placed in tubes with controlled humidity: a moistened, tightly packed cotton layer at the bottom, covered with filter paper, and sealed with a dry cotton-gauze plug.

We fully acknowledge the importance of sampling ticks from alternative hosts, particularly small mammals and migratory birds, as they play a crucial role in pathogen circulation and tick dispersal. Given the diverse ecosystems in the studied biotopes, where Ixodes persulcatus can feed on a wide range of mammals and birds, we plan to expand our monitoring efforts in future studies to include these hosts. This will provide a more comprehensive understanding of pathogen prevalence and tick ecology in the region.

Comments 2: Also, did you analyse only the Yezo virus, or were other pathogens included? It could be very interesting to determine if co-infections occur within the ticks (again, just a personal interest).

Response 2: Thank you for your insightful question. In addition to the Yezo virus, all collected ticks were screened for a broad range of pathogens commonly found in the Asian part of Russia. These included:

  • Viruses: Tick-borne encephalitis virus (TBEV), Kemerovo virus (KEMV), and segmented flavi-like viruses (Alongshan virus and Yanggou tick virus);
  • Bacteria: Borrelia spp., Rickettsia spp., and Anaplasma spp.;
  • Protozoa: Babesia canis.

This comprehensive screening allowed us to assess potential co-infections in ticks, which is indeed an important aspect of understanding pathogen transmission dynamics. Detailed results on pathogen prevalence and co-infection patterns, including their ecological and epidemiological implications, will be presented in a separate follow-up manuscript currently in preparation.

We fully agree that investigating pathogen interactions in ticks is a critical and promising research direction, and we appreciate the reviewer’s interest in this topic.

Comments 3: Figure 2 can be improved – does the representation of genome organisation only cover the coding regions? Because the figure shows three parts, 505, 1356, and 3938 bp, but according to the data in paragraph lines 187-195, the sizes do not match. Please provide a hint to the reader on how to interpret this figure.

Response 3: We sincerely appreciate your careful examination of Figure 2 and the opportunity to clarify this important aspect. You are absolutely correct in noting that the schematic representation specifically depicts only the coding regions of the Russian YEZV isolates' genome, as we have now made more explicit in the revised figure caption (line 198).

Comments 4: Figure S6 has really bad quality, please upload it in a better resolution.

Response 4: We sincerely apologize for the quality issues with Figure S6 in the original submission. The figure has now been replaced with a high-resolution version that maintains all necessary details for clear interpretation.

Comments 5: When it is possible, please add the Python script in the GIT hub (line 114), as it is also part of the method.

Response 5: Thank you for your suggestion regarding the Python script. We have now uploaded the script to a GitHub repository and included the direct link to it in the 2. Materials and Methods section (lines 115-116 in the revised manuscript). 
